# Remdesivir Treatment in Moderately Ill COVID-19 Patients: A Retrospective Single Center Study

**DOI:** 10.3390/jcm11175066

**Published:** 2022-08-29

**Authors:** Vedrana Terkes, Karla Lisica, Martina Marusic, Nikola Verunica, Anela Tolic, Miro Morovic

**Affiliations:** 1Department of Infectious Diseases, Zadar General Hospital, 23000 Zadar, Croatia; 2Emergency Department, Zadar General Hospital, 23000 Zadar, Croatia; 3Department of Cardiology, Zadar General Hospital, 23000 Zadar, Croatia; 4Department of Radiology, Zadar General Hospital, 23000 Zadar, Croatia

**Keywords:** remdesivir, COVID-19, early treatment, moderately ill patients, clinical outcome

## Abstract

Almost two years after remdesivir was approved and extensively used in numerous clinical studies for the treatment of COVID-19 patients, there is still no clear recommendation for the time and phase of the disease of remdesivir administration. This retrospective observational study included adults (≥18 years) with severe COVID-19, radiologically confirmed pneumonia, a need for supplemental oxygen and an interval from symptom onset to enrolment of 10 days or less. All patients were treated with remdesivir for 5 to 10 days, or with clinical improvement within that period. The primary goal was the outcome in patients treated with remdesivir during the early stage of the disease considering the different disease severity. The median time from symptom onset to treatment was 8.4 days (3–10). Clinical improvements and good outcomes were observed in 104 of 137 patients (75.9%); 33 (24.1%) of 137 patients died. Subgroup analyses showed that the mortality rate was significantly lower in moderately ill patients (3 out of 51 patients; 5.9%) than in the group of severely/critically ill patients (30 out of 86 patients; 34.8%; *p* < 0.005). Older age, rise of CRP and CT score were shown to be significant predictors of disease outcome. Overall, remdesivir was well tolerated, and the treatment was discontinued in only four patients. The results of this observational study in 137 patients with different disease severity contribute to the attitude concerning remdesivir administration in the early stage of COVID-19, at least in moderately ill patients with a high risk of progression, before the transition to a more severe stage.

## 1. Introduction

Remdesivir was the first Food and Drug Administration (FDA)-approved drug for the treatment of severe coronavirus disease 2019 (COVID-19) [1]. Recently, the FDA issued EUAs (Emergency Use Authorizations) that allow two new oral antiviral drugs to be used as treatments for COVID-19 in non-hospitalized patients with mild to moderate COVID-19 who are at high risk of progressing to serious disease: ritonavir-boosted nirmatrelvir and molnupiravir, but also remdesivir as an option, within 7 days of symptom onset [2]. Remdesivir is a nucleoside analog that acts as a competitive inhibitor of viral RNA-dependent RNA polymerase, with broad activity against many RNA viruses [3,4].

There are contradictory reports of remdesivir’s benefits in COVID-19 patients. On one hand, some randomized clinical trials showed no clinical benefit from remdesivir use [5,6]. A very recent, phase 3, randomized, controlled, open-label trial in 857 patients also showed no clinical benefit of remdesivir in patients who were admitted to hospital for COVID-19 within 7 days of the onset of symptoms, and who required oxygen support [7]. In addition, the WHO’s solidarity trial, which had as a primary goal the effect of treatment on in-hospital mortality, also showed that remdesivir had no effect on overall mortality [8].

On the other hand, a large, phase 3, double-blinded, placebo-controlled trial—the Adaptive COVID-19 Treatment Trial (ACTT-1)—showed that remdesivir was superior to a placebo in shortening the time to recovery in adults who were hospitalized with COVID-19 and produced evidence of lower respiratory tract infection; moreover, this study showed the larger benefit of remdesivir when given earlier, i.e., 10 days within the onset of symptoms than in those treated later [9]. Another randomized study also showed that remdesivir treatment was associated with significantly higher recovery rates and lower mortality than standard-of-care treatment without remdesivir in patients with severe COVID-19 [10]. 

However, the mentioned studies [5,6,7,8,9,10] did not clarify response to remdesivir when the disease severity categories or duration of symptoms before the treatment were stratification criteria. Therefore, the important questions, such as when and to whom to give remdesivir, remain open. In this article, the results of a retrospective study of remdesivir response in patients with COVID-19, stratified according to the disease severity criteria and to the duration of symptoms before the treatment, are presented.

## 2. Materials and Methods

This article retrospectively analyzed the data of 137 patients who were admitted to the tertiary care center in Zadar General Hospital, Croatia, between 21 June 2020 and 9 February 2021 and received remdesivir. Eligible patients were adult patients aged over the age of 18 years. They were real-time reverse transcription polymerase chain reaction (RT-PCR) positive for SARS-CoV-2, had radiologically confirmed pneumonia, a need for supplemental oxygen and were given remdesivir within 10 days of symptom onset. All the patients were grouped according to the disease severity criteria [2,11]. Exclusion criteria included pregnant or lactating women, hepatic cirrhosis or raised aminotransferases level greater than five times the normal upper limit and patients with severe renal impairment (estimated glomerular filtration rate < 30 mL/min/1.72 m^2^) or patients on dialysis. The initial evaluation included chest x-ray, electrocardiogram (ECG), complete blood count (CBC) with differential and metabolic profile, including liver and renal functional tests, C-reactive protein (CRP), D-dimer, ferritin, interleukin-6 (IL-6) and procalcitonin. Computed tomography (CT) was performed in 66 patients, mostly in the severe and critically ill (58 or 87.8%) and in a number (8 or 12.1%) of moderately ill patients with signs of disease progression. The CT score was calculated based on the extent of lobar involvement [2,11]. The patients were treated with 5- to 10-day courses of remdesivir, or to clinical improvement within that period. Remdesivir was administered as 200 mg intravenous infusion on day 1, followed by once daily, 1 h infusions of 100 mg. The day when saturation in room air significantly improved or supplemental oxygen was discontinued was defined as the day of clinical improvement. The vast majority of the patients received dexamethasone simultaneously (127; 92.7%), except in some cases of severe diabetes. All the patients received supportive care according to the standard of care (supplemental oxygen, low molecular weight heparin and dexamethason, except in a few cases). The research was approved by the Ethics Committee of Zadar General Hospital (under number 02-3673/21-9/21). All the patients gave verbal consent for the treatment since they were unable to give their written informed consent because of isolation precautions and the Ethics Committee waived the requirement. All investigations were conducted according to the principles expressed in the Declaration of Helsinki.

### Statistical Analysis

To answer the question of whether the sociodemographic variables of the patients predicted the outcome of the disease, a binary regression analysis was performed. Age, sex (0—women; 1—male) and presence of comorbidities (0—no comorbidities; 1—comorbidities) were used as predictors, and the criterion variable was disease outcome (0—death; 1—survival). The set model was statistically significant (X^2^ = 20.41; df = 3; *p* < 0.05) and the Nagelkerke R2 value was 0.207. The results of the significance of each predictor variable are shown in Table 4.

To test which of the biochemical as well as clinical parameters predict disease outcome, we performed a binary regression analysis with disease outcome as the criterion (0—death; 1—survival) and a number of biochemical parameters as predictors. The biochemical parameters were collected at two time points (first, on patients’ admission and second, during the remdesivir treatment). The tested model was statistically significant (X^2^ = 80.984; df = 14; *p* < 0.05) and the value of Nagelkerke R2 was 0.709. The results are shown in Table 5.

## 3. Results

The demographic and clinical characteristics of one hundred and thirty-seven hospitalized patients treated with remdesivir are shown in Table 1. The median time of hospitalization was 15 (1–59) days. The median age was 65 years (22–94); men accounted for 81% (111/137), mainly over 65 years (80/137, 58.4%). The most frequent coexistent disease was hypertension (80; 58.4%), followed by diabetes mellitus (44; 32.1%) and oncological disorders (14; 10.2%). Eighty-six (62.7%) patients were categorized as severe or critically ill, and the rest (51, 37.2%) as moderately ill. Chest radiographs revealed abnormal results in almost all patients. The most common finding was bilateral pneumonia (131; 95.6%) with the median CT score of 18.2 (8–25) in 66 (48.2%) patients (Figure 1).

The clinical course and outcomes of patients treated with remdesivir are shown in Table 2. During the observational period 24.1% (33/137) of patients died, mostly older than 65 years. Overall, 45 (32.8%) patients needed intensive care with high-flow oxygen use or mechanical ventilation. Clinical improvement and good outcome were observed in 75.9% (104/137) of patients with a median time to clinical improvement of 7.3 days. Remdesivir was generally well tolerated and the most common adverse events were not serious in about 20% of patients: nausea, headache and constipation; the most common laboratory abnormalities were hypokalemia, anemia and thrombocytopenia; in only four patients, remdesivir was discontinued because of a significantly elevated alanine aminotransferase level.

Clinical outcomes of patients treated with remdesivir are shown in Table 3. The mortality was significantly higher in the severe group (11/47 patients died; 23.4%) than in moderately ill patients (3/51 patients died; 5.9%; *p* = 0.0287; this difference was particularly expressed when moderately ill patients were compared with severe/critically ill patients (30/86 died, 34.8%, *p* = 0.0003). Moreover, the clinical output comparison of the 5-day (16/95 patients died; 16.8%) and 10-day (17/38 patients died; 44.7%) remdesivir course in our study showed a statistically significant difference (*p* = 0.0070). 

The influence of sociodemographic characteristics on the disease outcome in patients treated with remdesivir is presented in Table 4.

Table 4 shows that only patients’ age was a significant predictor of survival/death; older patients were more likely to die. Gender and the presence or absence of comorbidities were not significant predictors. In the cases with more comorbidities, there was also no statistically significant correlation with the outcome of the disease. On the basis of Nagelkerke R2, we can say that age can explain approximately 20% of the variance of the outcome of the disease.

The values of some biochemical and clinical parameters in patients treated with remdesivir are shown in Table 5.

Table 5 shows that the only two significant predictors of disease progression were CRP (measured at control) and CT score. A higher CRP2 score was a significant predictor of death, and a higher CT score was significantly related to a higher chance of death.

## 4. Discussion

A recent systematic study, which analyzed and evaluated the diagnosis and treatment guidelines for SARS-CoV-2 infection, showed that 22 from a total of 30 guidelines referred to antiviral therapy; in general, the use of these drugs has shown to be in most cases conditional [12]. For example, remdesivir has been widely used in many countries, with several guidelines recommending its use in patients with severe and critically ill COVID-19 [2,13]. The clinical course and outcomes regarding the time from onset of symptoms of COVID-19 to the initiation of remdesivir were analyzed in only a few clinical and experimental studies. In general, for some viral illnesses, such as influenza, the efficacy of antiviral administration has shown to be most effective when administered within 48 h after symptom onset [14]. Moreover, an argument in favor of the early use of remdesivir is produced from one in vitro investigation of oral ribonucleoside analog with broad-spectrum antiviral activity against various RNA viruses, which showed that the efficacy of direct-acting antivirals against acute viral respiratory infections typically decreased with delay in treatment initiation [15].

In an incomplete clinical study, Wang et al. reported that patients with severe COVID-19 who received remdesivir within 10 days from the onset of symptoms had a numerically faster time of clinical improvement (median 18.0 days), although not statistically significant, than those receiving a placebo (median 23.0 days); unfortunately, this trial was stopped early due to the lack of eligible patients and there was no conclusive result [5]. In this context, an exploratory analysis of the Gilead Phase 3 Trial of investigational antiviral remdesivir in patients with severe COVID-19 suggests that patients who received remdesivir within 10 days from symptom onset had improved outcomes compared with those treated after more than 10 days of symptoms [16]. Moreover, a retrospective study by Mehta et al. showed that in-hospital all-cause mortality was significantly lower in patients with moderate to severe COVID-19 who received remdesivir within 9 days from symptom onset than in those who were treated after that period [17]. Our results, which showed a short median time to clinical improvement (7.3 days), and a good outcome in more than two thirds of remdesivir-treated patients, favored the initiation of treatment in this early stage of COVID-19. However, transition into a more severe stage of the illness occurred in a significant proportion of our treated patients, as elevated markers of systemic inflammation suggested, and it is clear that the antiviral effect of remdesivir alone was not sufficient for disease control. As proposed, after the early stage of COVID-19, host inflammatory response ensued, and, therefore, additional treatment with anti-inflammatory agents was necessary. In this context, two recent studies must be pointed out. The first was treatment combination with remdesivir and baricitinib, an orally selective inhibitor of Janus kinase (JAK) 1 and 2, which proved to be superior to remdesivir alone in reducing the recovery time and accelerating improvement in the clinical status among patients with COVID-19 [18]. The second one showed that the combination of tocilizumab and remdesivir could be of benefit in severe COVID-19 patients, but also the combination of tocilizumab and hydroxychloroquinne [19]. Moreover, although our study was not part of a clinical trial (with a placebo or some other medication control), the finding of high mortality among patients who did not receive remdesivir, 129 out of 323 (40%) vs. 33 (24%) out of 137 treated patients must be outlined; these SOC-treated patients were admitted mostly after a period of 10 days from symptom onset, i.e., during the pulmonary and hyperinflammatory phases of the disease usually connected with more complications and progression to the acute respiratory distress syndrome (ARDS), but it must be noted that remdesivir was not available continuously during the observational period. In this regard, the mentioned comparative analysis of remdesivir vs. SOC treatment in adults with severe COVID-19 showed that remdesivir was associated with significantly greater recovery and a 62% reduction in the odds of death [9]. The mortality rate in our group of moderately ill patients with a high risk of progression was statistically significantly lower (3 out of 48 patients; 5.9%) when compared with the high mortality in the severely ill (11 out of 47 patients; 34.9%; *p* = 0.0287) ones.

A randomized study in moderately ill patients, with mean duration of symptoms before the first dose of 8 days, showed that those randomized to a 10-day course of remdesivir did not have a statistically significant difference in clinical status compared with standard care at 11 days after initiation of treatment, while patients randomized to a 5-day course of remdesivir had a statistically significant difference in clinical status compared with standard care, but the difference was of uncertain clinical importance [6]. The two open-label, multi-center Phase 3 Gilead SIMPLE trials, where patients were randomized to a 5- or 10-day course of remdesivir, also did not show a significant difference [16]. The clinical output comparison of the 5-day (16/95 patients died; 16.8%) and 10-day (17/38 patients died; 44.7%) remdesivir course in our study showed a statistically significant difference (*p* = 0.0070). However, all these analyses favor the use of the 5-day regimen to escape possible adverse events and unnecessary costs.

In comparison with the low overall mortality of 6.7% in the mentioned ACTT-1 study of remdesivir treatment of COVID-19 in 541 patients by day 15, and of 11.4% by day 29 [7], the mortality rate in our patients was three times higher (24.1%). A fatal outcome in our patients was seen almost exclusively among the severe and critically ill patients (30 from 86; 34.9%). The significant predictors of death in our group ofl 147 patients were older age, a singificante rise of CRP (Table 5) and a high CT score (a mean value of 18.2), while the influence of comorbidities was not confirmed.

A greater systematic review of forty-two studies showed that the pooled prevalence of mortality among 423,117 hospitalized patients with COVID-19 was 17.6%; this study showed that significant risk factors for death outcome are older age, male gender, chronic obstructive pulmonary disease (COPD), cardiovascular disease, diabetes, hypertension, obesity, cancer, acute kidney injury and increase D-dimer [20].

Our study had some limitations. The study population was conducted at a single center with a limited sample size and the results cannot be generalized. However, the new wave of SARS-CoV-2 variants is presently going on very intensively and it could be expected that results with the inclusion of many more patients will further clarify the benefit of remdesivir use. Moreover, it might be expected that remdesivir will retain activity against even new variants with adaptive mutations since it has a defined viral target and genetic barrier to resistance development [21,22,23].

## 5. Conclusions

When summarized, our results showed that the administration of remdesivir in the early stage of COVID-19 could be of benefit at least in a group of moderately ill patients, before the transition into a more severe stage of the disease. It is our opinion that remdesivir has its place in the treatment of COVID-19, but the defining of an appropriate time of the treatment initiation needs more than experimental, observational and case studies.

## Figures and Tables

**Figure 1 jcm-11-05066-f001:**
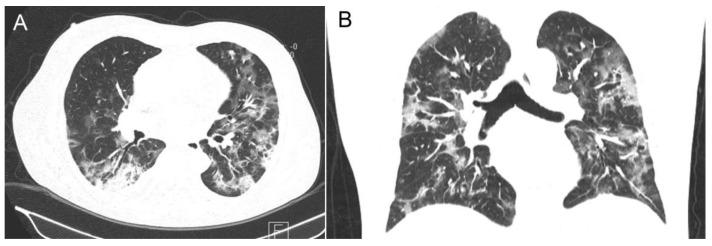
A male patient in early 50 s, categorized as a moderately ill COVID-19 patient, presenting symptoms for 13 days; chest CT scan shows multilobar, diffuse consolidation and ground glass opacification; some features of organizing pneumonia (CT score = 18). (**A**) axiale slice. (**B**) coronal slice.

**Table 1 jcm-11-05066-t001:** Demographic and clinical characteristics of 137 hospitalized patients treated with remdesivir.

Characteristics	
Age: median (range)—years	65 (22–94)
Sex—male, *n* (%)	111 (81)
Underlying disease, *n* (%)	
Hypertension	80 (58.4)
Diabetes	44 (32.1)
Hypertension and diabetes	30 (21.9)
Malignancy	14 (10.2)
State of illness, *n* (%)	
Moderate	51 (37.2)
Severe	47 (34.3)
Critical	39 (28.5)
Days from symptoms to treatment initiation, median (range)	8.4 (3–10)
Radiological findings	
X-ray, bilateral pneumonia, *n* (%)	131 (95.6)
CT-score; *n* (%); median; range	66 (48.2); 18.2 (8–25)

**Table 2 jcm-11-05066-t002:** Clinical course and outcomes of patients treated with remdesivir.

Course and Outcome	N (%)
Died	33 (24.1)
>65 years of age	25 (75.7)
Severe/critical (out of 86 patients)	30 (34.8)
Moderate	3 (5.9)
ICU care	45 (32.8)
Clinical improvement and discharged	104 (75.9)
Time to clinical improvement: median; No. of days	7.3; 3–25
Hospital stay: median; No. of days	15; 5–59
Adverse events (the most common)	
Nausea	34 (24.8)
Constipation	26 (18.9)
Headache	28 (20.4)
Hypokalemia	16 (11.6)
Anemia	15 (10.9)
Thrombocytopenia	15 (10.9)
Discontinued treatment	
Significantly increased alanine aminotransferase	4 (2.9)

**Table 3 jcm-11-05066-t003:** Death rate of patients treated with remdesivir.

Characteristics	Death No (%)	*p*-Value
Severity		
Moderate	3/51 (5.9)	0.0287 *
Severe	11/47 (23.4)	
Critical	19/39 (48.7)	0.0003 **
Duration of treatment		
5-day	16/95 (16.8)	0.0070
10-day	17/38 (44.7)	

* Moderate vs. severe. ** Moderate vs. severe/critical.

**Table 4 jcm-11-05066-t004:** Prediction of disease outcome based on sociodemographic variables.

Variable	Estimate	Standard Error	*p*-Value	95% Confidence Interval	
				Lower	Upper
Gender	−0.311	0.561	>0.05	0.579	0.733
Age	−0.076	0.021	<0.05	0.000	0.926
Comorbidity	−0.413	0.580	>0.05	0.476	0.661

Gender: 0—male, 1—female; Comorbidity: 0—no, 1—yes; Disease outcome: 0—death, 1—survival.

**Table 5 jcm-11-05066-t005:** Biochemical and clinical variables as predictors of disease outcome (0—death, 1—survival).

* Variable	Estimate	Standard Error	*p*-Value	95% Confidence Interval	
				Lower	Upper
L1	−0.083	0.082	>0.05	0.784	1.079
L2	0.078	0.082	>0.05	0.921	1.269
Ly1	0.043	0.094	>0.05	0.868	1.254
Ly2	0.071	0.055	>0.05	0.964	1.195
NLR1	0.026	0.041	>0.05	0.947	1.113
NLR2	−0.003	0.033	>0.05	0.934	1.064
CRP1	0.000	0.004	>0.05	0.992	1.008
CRP2	−0.055	0.007	<0.05	0.965	0.992
PCT1	−0.005	0.052	>0.05	0.899	1.103
PCT2	−0.129	0.408	>0.05	0.395	1.955
D.dim1	−0.031	0.037	>0.05	0.902	1.043
D.dim2	−0.093	0.059	>0.05	0.811	1.024
CT score	0.091	0.044	<0.05	1.004	1.194

* L1—lymphocyte number at admission; L2—lymhocyte number at control during the treatment; Ly1—lymhocyte number at admission; Ly2—lymphocyte number at control; CRP1—at admission; CRP2—at control; D-Dimer 1—at admission; D-Dimer 2—at control.

## Data Availability

Data available upon request from the corresponding author.

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
