# Peer review of "Remdesivir Treatment in Moderately Ill COVID-19 Patients: A Retrospective Single Center Study"

_jcm, 2022, doi:10.3390/jcm11175066_

Round 1

Reviewer 1 Report (Previous Reviewer 1)

The authors have revised the manuscript (jcm-1732886 V2 to jcm-1895145 V1). Therefore, the author did not thoroughly check the manuscript, including the writing of COVID-19. COVID-19 and COVID-19 are both written in the manuscript. Furthermore, please double-check the sentences in L 191 that include space and a comma, as well as the font size in 244-257.

Author Response

We revised manuscript according to your request.

Reviewer 2 Report (Previous Reviewer 2)

The paper was significantly improved. 

Author Response

Thank you for your patience.

Reviewer 3 Report (Previous Reviewer 3)

I have reviewed the new version of the article, which is much improved. I made some small corrections directly into the text.

It needs revision of certain phrases especially from the English language point of view.

Author Response

We revised manuscript according to your request.

This manuscript is a resubmission of an earlier submission. The following is a list of the peer review reports and author responses from that submission.

Round 1

Reviewer 1 Report

Dear Authors,

Please see the attachment below.

Author Response

Ad 1. Correction of English language is done

Ad 2. Word „we“ is remowed from all sentences.

Ad 3. All abbrevations are defined in the first sentence.

Ad 4. Explaining the ACCT-1 study.

Ad 5. Citation on L 30-31 included.

Ad 6. Covid -19 is corrected.

Ad 7. L-39 rhe space is fixed

Ad 8. In reference 7 (in new version 8) te date of access is included.

Ad 9. L 48-51 is renewed with reference.

Ad 10. The remdesivir dose is inccluded in the method section.

Ad 11. Dots are corrected.

Ad 12. The Ethic Number from the Zadar General Hospital Ethics Committee is included in the text.

Ad 13. Capitalization of „hypertension and diabetes“ in Table 1 are corrected.

Ad 14. The differentiation for Diabetes types was not provided, but almost all of the diabetic patients were not insulin dependent.

Ad 15. Lines 91-93 are correlated with the data in Figure 1.

Ad 16. L 108-L111: the remdesivir dose was according to standard protocol ( loadin dose 200 of remdesivir on day 1 and 100 mg once daily thereafter) and there was no comaprison with other studies.

Ad 17. The spaces betwen L 113 and 114, as well as in L 115 are removed

Ad 18. Text in Table 2 is corrected.

Ad 19.  Text was shortened (L154-L168) and appropirate references are included.

Ad 20. Hydroxychloroguine20....and others mentioned were misprints.

Ad 21. Subject „we“ has been avoid during the text

Ad 22. Standard care is explained under the Material and Methods section. Ttatistically significance is included.

Ad 23.  The text is shortened and corrected.

Ad 24.  This is a misprint, there is only Figure 1, divided in A and B pictures

Ad 25. Conclusion is revisited.

Reviewer 2 Report

The manuscript by Terkes et al. evaluated the impact of remdesivir on outcome of COVID-19 infected individuals with different severity of disease

Some points need to be discussed

1) Larger trials did not confirm any impact of remdesivir on the mortality of COVID-19 infected individuals; it has been associated with a reduction of positivity but non with a reduced mortality. This study found opposite results, even if there is not a definite control group that did not receive the drug. This finding is completely misleading to me

2) To determine that remdesivir is useful in the early stage of COVID-19 disease authors should compare patients who started treatment < 5 days after symptoms onset with others. If all the patients started remdesivir within 10 days from the symptoms initiation how can they state the efficacy in the early disease?

3) It's not clear how the investigators decided if a patient received or not received remdesivir. Who decided? Which criteria were considered? And further, how they decided to treat patients for 5 or 10 days?

4) Authors described that 99 patients received dexamethasone. They stated that this is 93,4% but the overall number of patients is 137. Please reconsider the proportion

5) Which dosage of dexamethasone was used? And which was the duration of steroid treatment?

6) How they defined the severity of disease? They used a score? Or a clinical criterium? Please specify 

7) Most of patients had a need for oxygen supplementation. How many patients underwent to invasive or not invasive ventilation? The presence of a large number of patients needed invasive ventilation may explain the high rate of mortality observed

8) The patients who died had other kind of complications leading to death? For example cardiac or infective complications?

9) The discussion should be shortened; in the present form it seems a review of the literature. Authors should focused more on their results

10) A lot of typing errors are present in the discussion section (some numbers after sentences, as at line 180, 190 or 210)

Author Response

Ad 1. This study extracted of moderately ill COVID-19 patients with a high risk of progression treated with remdesivir as a group with significantly lower mortality in comparison to severely ill treated patients.

Ad 2.  We accepted the 3-stage classification system of COVID-19 illness with 3 grades of increasing severity, corresponding clinical findings and response to therapy (Siddiqui HK, 2020), in which antiviral therapy as remdesivir in the early stage I  (early infection) could prevent progression of the disease. In this regard eligible patients in the study were those who received remdesivir within 10 days of symptoms onset and we did not compare patients treated before or after 5 days from symptoms intiation. However, it is of interest to say that 35 from 137 patients (25.5%) received remdesivir within 5 days of symptoms onset, with mortality rate of 34,3% (12/35) and all of them belonged to the severe/critical ill groups.

Ad 3. As we stated in the text, remdesivir was not available continuously during the observational period. Basically, the treatment was not planned as a 5- or 10-days defined treatment protocol and its duration depended on the clinical course, i.e. if the clinical improvement ensued earlier the treatment stopped earlier.

Ad 4. This was a misprint and the numbers and proportion are corrected in the text, i.e. 127 from 137 patients (92.7%) received dexamethason.

Ad 5. The dose of dexamethason was 6 mg daily up to 10 days with the mean duration of treatment of 8 days.

Ad 6. As stated under the Materials and Methods section criteria of the disease severity was in accordance NIH COVID-19 guidelines (ref.2).

Ad 7. Invasive ventilation was applied in 39 from 137 patients, non invasive in 98 patients. The mortality rate was explained in the Discussion section in this regard.

Ad. 8. In the patients who died, in 13/33 elevated CRP (>150 mg/L) was registered, while in 7/33 characteristic radiological signs of  bacterial pneumonia was found; in one patient thromboembolic changes of a leg was seen and he underwent to mputation, in one patient pneumotorac and pneumomediastinum was seen.

Ad 9. The discussion is shortened,

Ad 10.Typing errors are corrected.

Reviewer 3 Report

The article is interesting, but the approach is one-sided and somewhat simplistic, only from the point of view of the infectionist. An interesting clinical experience is presented only from a statistical point of view.

Although the association of comorbidities such as hypertension, diabetes mellitus, including also the combination, as well as neoplasms is mentioned, it is not clear what the impact of the treatment was in patients with comorbidities, nor about the particular evolution of these cases.

Certainly the patients who associated hypertension and diabetes also had heart damage and it would have been interesting to discuss the evolution of these cases with comorbidities compared to cases without cardiovascular comorbidities.

 The small number of included patients in the study is not relevant.

Author Response

The basic idea of our retrospective observational study was to investigate is it possible to affect the viral replication during the early COVID-19 infection (stage I) and prevent the disease progression with antiviral drug remdesivir in patients with various degree of disease severity.

The influence of comorbidities such as hypertension, diabetes mellitus and neoplasms were not separately analysed because extensive reports were published elsewhere (Guan WJ et al. 2020, Ejaz H et al. 2020, Li J et al  2021, Eč-Badawy O et al.2022.). The number of the patients was related to the observational period we choosed (2020-2021).

Round 2

Reviewer 1 Report

The authors have revised the manuscript sufficiently in response to my comments. However, the flow remains challenging to follow. I can therefore recommend publication after a thorough English editing by professionals, including paraphrasing. Additionally, please check the spelling of COVID-19. Covid-19 or COVID-19? (Lines 41, 51, 44, etc.)

Author Response

The Introduction section and belonging references are adapted according to reviewer's suggestion.

The results are better presented with the introduction of subgroups of died patients and their comorbidities into the text.

Our English is revisited by a professional.

Reviewer 2 Report

The paper presents severe methodological biases that were not significantly improved after revision. These biases are mainly due to the retrospective method of analysis and unfortunately can significantly affect the results provided.

To assess death rate, a multivariate analysis is more appropriate tthan an univariate, to better evaluate the role of different predictors.

The high rate of mortality observed should be analyzed deeply in order to find predictors for that (age? gender? co-morbidities?, etc)

Among co-morbidities I would consider also other diseases if known (auto-immune?, concomiztant infection?)

In the discussion, the hypothesis of a more virulent variant is just hypothetical without a specific analysis. This comment in my opinion should be deleted

Author Response

The Introduction section is improved according to reviewer's suggestion.

The explanation of high mortality rate is reconsidered by including the died patients' subroups and their comorbidities.

The multivariate analysis was not done due to the very small sample and no control group(s) to compare.

The hypothesis of a more virulent variant is thrown out.

Reviewer 3 Report

I have read the new version of the article and I have noticed the corrections and the explanations to my previous observations.

The article was improved in this new version

Author Response

The article was improved in this previous version.